# Technical Note: Influence of surface roughness and local turbulence on coated-wall flow tube experiments for gas uptake and kinetic studies

Guo Li[1,4], Hang Su[2,1*], Uwe Kuhn[1], Hannah Meusel[1], Markus Ammann[3], Min Shao[2,4], Ulrich Pöschl[1],

Yafang Cheng[1,2*]

[1] Multiphase Chemistry Department, Max Planck Institute for Chemistry, Mainz, Germany

[2] Institute for Environmental and Climate Research, Jinan University, Guangzhou, China

[3] Laboratory of Environmental Chemistry, Paul Scherrer Institute, 5232 Villigen, Switzerland

[4] College of Environmental Sciences and Engineering, Peking University, Beijing, China

\* *Correspondence to*: Y. Cheng (yafang.cheng@mpic.de) or H. Su (h.su@mpic.de)

**Abstract**

Coated-wall flow tube reactors are frequently used to investigate gas uptake and heterogeneous or multiphase reaction kinetics under laminar flow conditions. Coating surface roughness may potentially distort the laminar flow pattern, induce turbulence and introduce uncertainties in the calculated uptake coefficient based on molecular diffusion assumptions (e.g., Brown/CKD/KPS methods), which has not been fully resolved in earlier studies. Here we investigate the influence of surface roughness and local turbulence on coated-wall flow tube experiments for gas uptake and kinetic studies. According to laminar boundary theory and considering the specific flow conditions in a coated-wall flow tube, we derive and propose a critical height $\delta_c$ to evaluate turbulence effects in the design and analysis of coated-wall flow tube experiments. If a geometric coating thickness $\delta_g$ is larger than $\delta_c$, the roughness elements of the coating may cause local turbulence and result in overestimation of the real uptake coefficient ($\gamma$). We further develop modified CKD/KPS methods (i.e., CKD-LT/KPS-LT) to account for roughness-induced local turbulence effects. By combination of the original methods and their modified versions, the maximum error range of $\gamma_{CKD}$ (derived with the CKD method) or $\gamma_{KPS}$ (derived with the KPS method) can be quantified and finally $\gamma$ be constrained. When turbulence is generated, $\gamma_{CKD}$ or $\gamma_{KPS}$ can bear large difference compared to $\gamma$. Their difference becomes less for gas reactants with lower uptake (i.e., smaller $\gamma$), or/and for a smaller ratio of the geometric coating thickness to the flow tube radius ($\delta_g/R_0$). On the other hand, the critical height $\delta_c$ can also be adjusted by optimizing flow tube configurations and operating conditions (i.e., tube diameter, length and flow velocity), to ensure not only unaffected laminar flow patterns but also other specific requirements for an individual flow tube experiment. We use coating thickness values from previous coated-wall flow tube studies to assess potential roughness effects using the $\delta_c$ criterion. In most studies, the coating thickness was sufficiently small to avoid complications, but some may have been influenced by surface roughness and local turbulence effects.

## 1. Motivation

Coated-wall flow tube reactors have been extensively employed for investigations of uptake and reaction kinetics of gases with reactive liquid/semi-solid/solid surfaces (Howard, 1979;Kolb et al., 2010). To simulate various heterogeneous or multiphase reactions relevant to atmospheric chemistry, these coated reactive surfaces can span a broad scale including aqueous inorganic acids (Jayne et al., 1997;Pöschl et al., 1998), inorganic salts (Davies and Cox, 1998;Chu et al., 2002;Qiu et al., 2011), organic acids and sugars (Shiraiwa et al., 2012;Steimer et al., 2015), proteins (Shiraiwa et al., 2011), soot (McCabe and Abbatt, 2009;Khalizov et al., 2010;Monge et al., 2010), mineral dust (El Zein and Bedjanian, 2012;Bedjanian et al., 2013), ice (Fernandez et al., 2005;McNeill et al., 2006;Petitjean et al., 2009;Symington et al., 2012;Hynes et al., 2002;Hynes et al., 2001;Bartels-Rausch et al., 2005) and soils (Stemmler et al., 2006;Wang et al., 2012;Donaldson et al., 2014a;Donaldson et al., 2014b;VandenBoer et al., 2015;Li et al., 2016). Reactive uptake kinetics to a condensed phase material is normally described in terms of the uptake coefficient, $\gamma$, which represents the net loss rate of a gas reactant at the surface normalized to its gas kinetic collision rate. Due to uptake or chemical reactions of gases at the walls, radial concentration gradients can develop in the tube and radial diffusion can limit the observed gas uptake. The most commonly utilized methods for evaluating and correcting gas diffusion effects in flow tube studies include the numerical methods of Brown (Brown, 1978), and Cooney-Kim-Davis (CKD, Cooney et al., 1974;Murphy and Fahey, 1987;Davis, 1973) and the recently developed analytical Knopf-Pöschl-Shiraiwa method (KPS, Knopf et al., 2015). All of these methods are derived based on the assumptions that loss at the walls occurs through a first-order process (characterized by $\gamma$), and that the gas flow in flow tubes is a well-developed laminar flow. The second assumption ensures that the flow velocity profile is parabolic and that the radial transport of the gas reactant is solely caused by molecular diffusion.

It is well known that the flow conditions in a tube depend on the Reynolds number, $Re$ (Eqn. 1),

$$Re = \frac{\rho \times V_{avg} \times d}{\mu} = \frac{V_{avg} \times d}{v}$$

(1)

where $\rho$ is density of the fluid passing through the tube, $V_{avg}$ is average velocity of the fluid (i.e., the volumetric flow rate divided by the cross sectional area of the tube), $d$ is diameter of the tube, $\mu$ and $v$ are dynamic viscosity and kinematic viscosity of the fluid, respectively. A laminar flow can be expected when $Re$ is less than ~ 2000 (Murphy and Fahey, 1987;Knopf et al., 2015). Here, the expression of $Re$ quantifies the nature of the fluid itself (i.e., $\rho$, $V_{avg}$, $\mu$ and $v$) and the tube geometry (i.e., $d$), but it does not account for the effects of surface roughness. For a list of abbreviations and symbols used in the context see Appendix A.

Surface roughness effects on flow conditions were firstly discussed by Nikuradse (1950). Based on his work, the Moody diagram has been extensively used in industry to predict the effects of surface roughness (roughness height $\delta_r$ or relative roughness $\delta_r/d$) on flow characteristics (in terms of friction factor). According to the Moody chart, when the surface roughness is small enough (i.e., $\delta_r/d \leq 5\%$), the roughness effects within low Reynolds number regime ($Re < 2000$, characteristic of laminar flow) is negligible. Recent experimental and theoretical studies, however, have found significant effects of surface roughness on laminar flow characteristics (e.g., fraction factor, pressure drop, critical Reynolds number and heat transfer, etc.) in micro-channels and pipes even under conditions of $\delta_r/d \leq 5\%$ (Herwig et al., 2008;Zhang et al., 2010;Zhou and Yao, 2011;Gloss and Herwig, 2010). This is because not only the ratio of $\delta_r$ and $d$ but also other factors, such as shape of roughness elements (Herwig et al., 2008;Zhang et al., 2010) and spacing between different roughness elements (Zhang et al., 2010), may determine the influence of surface roughness on the flow conditions.

Moreover, compared to the rough pipe surfaces commonly dealt with in industry (with $0 \leq \delta_r \leq 50$ μm, see http://mdmetric.com/tech/surfruff.htm), the surfaces used in atmospherically relevant flow tube studies are with much larger surface roughness (e.g., inorganic salts, organic acids and proteins, soot, mineral dust, ice and soils, with $0 \leq \delta_r \leq$ ~ 650 μm; see Fig. 1), and the roughness of these surfaces are sometimes beyond the criterion of $\delta_r/d \leq$ ~ 5%. The reported specific surface areas of these coatings span a wide range from ~ 20 $m^2$ $g^{-1}$ to ~ 100 $m^2$ $g^{-1}$ with a coated film thickness scale from tens of micro-meters to several hundreds of micro-meters (Davies and Cox, 1998;Chu et al., 2002;McCabe and Abbatt, 2009;Khalizov et al., 2010;El Zein and Bedjanian, 2012;Bedjanian et al., 2013;Shiraiwa et al., 2012;Wang et al., 2012;Donaldson et al., 2014a;Donaldson et al., 2014b;VandenBoer et al., 2015). These geometrical characteristics indicate considerable porosity in coating layer and significant roughness on their surfaces.

Although the surface roughness effects can be potentially important, there has been a long-lasting debate on whether the coating surface roughness could disturb the fully developed laminar flow in flow tube kinetic experiments (Taylor et al., 2006;Herwig et al., 2008) and its effects were usually not well-quantified in most of the previous gas uptake or/and kinetic studies (Davies and Cox, 1998;Chu et al., 2002;McCabe and Abbatt, 2009;Khalizov et al., 2010;El Zein and Bedjanian, 2012;Bedjanian et al., 2013;Shiraiwa et al., 2012;Wang et al., 2012;Donaldson et al., 2014a;Donaldson et al., 2014b;VandenBoer et al., 2015;Li et al., 2016). It is, however, conceivable that as the roughness of the coating surfaces increases it would eventually distort the steady laminar regime near tube walls and small-scale eddies would evolve from roughness elements. These roughness-induced eddies will give rise to local turbulence, and hence corrupt the application of Brown/CKD/KPS methods for the correction of gas molecular diffusion effects and the determination of the uptake coefficient. The extent of these effects may depend on the coated film thickness and its surface roughness. It means that the roughness effects on flow conditions to a great extent rely on the various coating techniques applied by different operators, leading to disagreement of the experimental results.

In the present study, the surface roughness effects on laminar flow are quantitatively examined. In view of the special laminar boundary layer structure in flow tubes, we employ a critical height $\delta_c$, which defines the smallest scale within which local turbulence can occur (i.e., for scales smaller than $\delta_c$, local turbulence cannot exist, see Kolmogorov (1991)), to evaluate the influence of surface roughness on laminar flow patterns. By taking it into account in flow tube experimental design, it is feasible to satisfy the preconditions of merely radial molecular diffusion of gas reactants, and therefore validate the application of Brown/CKD/KPS methods. The $\delta_c$ criterion provides an easy way of assessing and optimizing different flow tube configurations and operating conditions (e.g., tube diameter, tube length, flow velocity, coating thickness, etc.) with regard to (1) the applicability and validity of diffusion correction methods, and (2) the specific requirements of an individual flow tube experiment design. To illustrate the applicability of the $\delta_c$ criterion, we analyze and assess previous coated-wall flow tube studies with regard to potential roughness effects. Moreover, we develop modified CKD/KPS methods accounting for the maximum impact of local turbulence (CKD-LT/KPS-LT) to assess how much the real uptake coefficient may deviate from the value obtained with the original CKD/KPS methods assuming purely molecular diffusion.

## 2. Methods

### 2.1 Influence of surface roughness on laminar flow

According to the proverbial boundary layer theory proposed by Prandtl (1904), when a fluid (normally a gas mixture, a gas reactant mixed with a carrier gas, in uptake kinetic studies) enters the inlet of a flow tube with a uniform velocity, a laminar boundary layer (i.e., velocity boundary layer) will form very close to the tube wall (Fig. 2). This buildup of laminar boundary layer is because of the non-slip condition of the tube wall and the viscosity of the fluid, that is, viscous shearing forces between fluid layers are felt and dominant within the laminar boundary layer (Mauri, 2015). The thickness of laminar boundary layer $\delta_l$ will continuously increase in the flow direction (axial direction in Fig. 2) until at a distance (from the tube entrance) where the boundary layers merge. Beyond this distance the tube flow is entirely viscous, and the axial velocity adjusts slightly further until the velocity along the axial direction doesn't change anymore. Then, a fully developed parabolic velocity profile is formed, characteristic of well-developed laminar flow (Mohanty and Asthana, 1979;White, 1998). The development and formation of this velocity profile is illustrated in Fig. 2. Normally, for coated-wall flow tube experiments a chemically inert entrance region with smooth surface is designed to ensure the development of laminar flow before the reactive gas enters into the coated-wall region.

As demonstrated in previous studies using micro-channels and pipes (Herwig et al., 2008;Gloss and Herwig, 2010;Zhang et al., 2010;Zhou and Yao, 2011), the roughness elements on flow tube coatings can have non-ignorable effects on laminar flow conditions even if these coatings are entirely submerged into the laminar boundary layer. In other words, the disturbance on well-developed laminar flow patterns can be artificially achieved by roughness elements of the tube coating. However, there is a critical height $\delta_c$ within which the roughness effects can become ignorable (Achdou et al., 1998).

Figure 3 shows a schematic of the structure of the $\delta_c$ and its related flow conditions in a coated-wall flow tube. When a roughness height $\delta_r$ (here in Fig. 3, the roughness height $\delta_r$ equates to the geometric coating thickness $\delta_g$, see Sect. 2.3 for explanation) comes into the critical height $\delta_c$ where viscous effects overwhelmingly dominate, the flow very near the rough wall will tend to be Stokes-like or creeping, denoted as laminar flow (LF) regime in Fig. 3A. This Stokes-like flow adjacent to the rough surfaces can avoid local turbulence between the roughness elements and guarantee perfect laminar flow conditions (i.e., only molecular diffusional transport of gas reactants to rough reactive coatings) throughout the whole flow tube volume. Thus LF regime satisfies the prerequisite for the diffusion correction methods used for flow tube experiments, i.e., $\delta_r/\delta_c < 1$. Nevertheless, when a roughness height is larger than the critical height $\delta_c$, local eddies may occur in the spaces between the neighboring roughness elements (i.e., local turbulence (LT) regime in Fig. 3B). Local turbulence induced by these roughness elements will enhance local transport of air masses within the scales of the roughness heights, which invalidates the assumption of solely molecular diffusion of gas reactants and therefore the application of diffusion correction methods for the determination of $\gamma$ (Brown, 1978;Murphy and Fahey, 1987;Knopf et al., 2015). In the next section, we will show how to derive $\delta_c$.

## 2.2 $\delta_c$ derivation

Achdou et al., (1998) proposed effective boundary conditions for a laminar flow over a rough wall with periodic roughness elements, and observed that when $\delta_r/L_c < Re^{-1/2}$ ($\delta_r$: roughness height; $L_c$: characteristic length, for a tube the characteristic length $L_c = d$) the roughness elements could be contained in the boundary layer. This means that, for their case, the boundary layer thickness is in the order of $L_c Re^{-1/2}$. Within the boundary layer, they found that local turbulence could occur between the roughness elements until $\delta_r/L_c < Re^{-3/4}$, where the viscous effects became dominated in roughness elements and then the flow near the rough wall tended to be creeping. This result coincides with Kolmogorov's theory (Kolmogorov, 1991), in which the critical length ratios between small scale and large scale eddies are also in the order of $Re^{-3/4}$, even though this theory only

applies to turbulent flow with large Reynolds numbers. Here, we adopt this criterion to judge if local eddies could occur in the spaces between neighboring roughness elements. Thus, the critical height $\delta_c$ can be expressed as:

$$\delta_c = d \times Re^{-3/4} = d^{1/4} \times \left( \frac{V_{avg}}{v} \right)^{-3/4}$$

(2)

where $d$ is diameter of the flow tube, $Re$ is the Reynolds number, $V_{avg}$ and $v$ are average velocity and kinematic viscosity of the fluid, respectively.

With Eqn. (2), for a specified experiment configuration (i.e., flow tube diameter, flow velocity and fluid properties) the critical height $\delta_c$ can be determined, and therefore the effects of coating roughness on laminar flow can be estimated provided the roughness height $\delta_r$ is known.

**2.3 Error estimation with modified CKD/KPS methods**

The potential effects of coating roughness on laminar flow are described and classified into two regimes in Fig. 3 (Sect. 2.1), in which only LF regime provides the ideal precondition ensuring that the diffusion correction methods (Brown/CKD/KPS methods) can be applied to obtain accurate $\gamma$ through flow tube experiments. Regarding LT regime, however, the roughness-induced effects can be quantitatively simulated, because local turbulence is constrained into the scale of the roughness height $\delta_r$ (Oke et al., 2017).

Hence, for LT regime, in order to estimate the potential error of the uptake coefficient derived from molecular-diffusion-correction using the conventional CKD/KPS methods, we further develop modified CKD/KPS methods (denoted as CKD-LT/KPS-LT, illustrated in Fig. 4) to account for local turbulence impact. In the CKD-LT/KPS-LT methods, some basic assumptions are made: (1) the scale of a roughness element is much larger than the size of pores inside the bulk coating and the macroscopic diffusion inside pores is not the domain of roughness-induced local eddies; (2) half of the surface roughness height is defined as the local-eddies-occurring region (i.e., $0.5\delta_r = R_m - R_g$); (3) the turbulent diffusion coefficient within the local-eddies-occurring region is infinitely large (i.e., the turbulent transport within it is extremely fast). When a coating is smooth, the mass-based coating thickness $\delta_m$ is equal to the geometric coating thickness $\delta_g$. In this case, the radial molecular diffusion distance from the tube centreline is $R_m$. While with large surface roughness height, the radial molecular diffusion distance is reduced to $R_g$. With the CKD-LT/KPS-LT methods, derivation of the uptake coefficient using $R_g$ rather than $R_m$ reflects an upper limit for the influence of local turbulence, as the turbulent diffusion coefficient in the local-eddies-occurring region is assumed to be infinitely large and turbulent transport occupies its whole volume. More details about CKD and KPS, and the derivations of $\gamma_{CKD}$, $\gamma_{KPS}$ $\gamma_{CKD\text{-}LT}$, and $\gamma_{KPS\text{-}LT}$ can be found in Appendix C-E.

**3. Results and discussion**

**3.1 Design of coated-wall flow tube experiments**

The introduction of the critical height $\delta_c$, into the field of gas uptake or reaction kinetic studies using coated-wall flow tubes, provides us the way for determining when the surface roughness effects can be negligible in flow tube experiments. That is, the roughness height $\delta_r$ of a coating film should be well within the domain of $\delta_c$ (LF regime in Fig. 3A). Only in this case, the free

molecular diffusion of a gas reactant in the radial direction can be ascertained and thus the Brown/CKD/KPS methods can be safely applied. Note that in real operations of flow tube coating design,several techniques (e.g., stylus profiler, non-contact optical profiler, scanning electron microscopy and atomic force microscopy, etc.) are available for surface roughness examination (Poon and Bhushan, 1995). To simplify the discussion, here, we take the geometric thickness of a coating film $\delta_g$ as a maximum of its surface roughness, and use the comparison between $\delta_g$ and $\delta_c$ as a reference for the design of flow tube coating thickness. Such treatment is more suitable for practical applications, because determination of coating film thicknesses can be simply achieved either by weighing the coating film mass (i.e., mass-based coating thickness $\delta_m$) or by utilizing scanning electron microscopy technique (i.e., geometric coating thickness $\delta_g$), and the condition of $\delta_g/\delta_c < 1$ can definitely ensure the case of $\delta_r/\delta_c < 1$. As discussed in Sect. 2.3, for coatings with large surface roughness their $\delta_g$ may be significantly larger than $\delta_m$. In this case, the criterion of $\delta_g/\delta_c < 1$ is more appropriate to be adopted.

Figure 5 shows the calculated $\delta_c$, with varying the tube diameter $d$ and the average flow velocity $V_{avg}$. From Eqn. (2), kinematic viscosity of a fluid (carrier gas in flow tubes) will affect $\delta_c$. It is therefore necessary to classify the flow tube experiments according to the types of the utilized carrier gases, such as synthetic air (Fig. 5A), nitrogen (Fig. 5B) and helium (Fig. 5C). For future flow tube coating design, Fig. 5 can be used to eliminate the potential coating surface roughness effects. Figure 6 summarizes and evaluates the potential effects of surface roughness in previous flow tube experiments. To reflect the influence of inherent roughness of the inner surface of a flow tube wall itself, the mean wall roughness is also accounted for coating thickness calculation when using rough-wall flow tubes (e.g., sandblasted tubes), for example, in the protein coating experiment. As shown in Fig. 6, most of the coating thicknesses are well below the calculated values of $\delta_c$ (LF regime), implying that their surface roughness effects on laminar flow and on the calculated uptake coefficient are ignorable. A few coating thicknesses, however, are significantly larger than the calculated $\delta_c$ (LT regime), as shown by the solid symbols. As the thicknesses of these two coatings are reported in terms of geometric coating thickness $\delta_g$ (Li et al., 2016;McNeill et al., 2006), they may have had a potential influence on laminar flow pattern and local turbulence may have occurred within the roughness-constructed spaces.

For most cases of flow tube experiments design, a coating layer cannot be thin enough due to requirements of reaction kinetics (bulk diffusion and surface reactions can both play important roles) and the thickness of a coating layer had been found to have an influence on gases uptake until a critical threshold was reached (Donaldson et al., 2014a;Li et al., 2016). This means that there is a need to comprehensively consider all the parameters (e.g., coating thickness, tube diameter, tube length, flow velocity, etc.) and a compromise of each parameter for the others is necessary to finally ensure both the unaffected laminar flow conditions and the specific requirements for an individual flow tube design. Larger $\delta_c$ would allow a wider range of coating thickness $\delta_g$ without surface roughness effects. Based on Eqn. (2), larger $\delta_c$ can be achieved either by increasing the tube diameter $d$ or by decreasing the fluid average velocity $V_{avg}$. Under the conditions of fast uptake kinetics, relatively short residence time of gas reactants inside coated-wall region is needed to allow for distinguishable penetration $C/C_0$ (i.e., the flow tube outlet concentration divided by the inlet concentration, see Fig. A2 for details). This requirement can be fulfilled by optimizing flow tube design. One can increase $d$ or decrease $V_{avg}$ to achieve larger $\delta_c$, but this operation will inevitably extend the residence time of gas reactants. Then, this effect can be offset by reducing $L$, which could be easily achieved by adjusting the position of a movable injector inside flow tube apparatus as in previous studies (Howard, 1979;Jayne et al., 1997;Pöschl et al., 1998;Kolb et al., 2010;VandenBoer et al., 2015).

**3.2 Divergences between different types of uptake coefficient due to molecular diffusion and local turbulence effects**

Normally, through coated-wall flow tube experiments, a penetration $C/C_0$ can be measured and therefore an effective uptake coefficient $\gamma_{eff}$ be experimentally determined (see Eqn. (C1) in Appendix C) under the assumption that the loss process on the wall is first-order. As discussed above, without roughness-induced local turbulence, the radial concentration gradient can give rise to molecular diffusion limitations of the gas reactant, which needs to be corrected using the diffusion correction methods (i.e., Brown/CKD/KPS) to derive the real uptake coefficient $\gamma$. Thus, the deviation between $\gamma_{eff}$ and $\gamma$ is only caused by molecular diffusion effects under ideal laminar flow conditions (LF regime).

With roughness-induced local turbulence (LT regime), the preconditions of conventional molecular diffusion correction methods can be corrupted. Figure 7 displays a schematic of different types of uptake coefficients and their divergences due to molecular diffusion and local turbulence effects. For LT regime, the conventional CKD or KPS may cause overcorrection of $\gamma_{eff}$, i.e., $\gamma_{CKD} \geq \gamma$ or $\gamma_{KPS} \geq \gamma$ (upper limit indicated in red in Fig. 7). In this case, the derived $\gamma_{CKD\text{-}LT}$ or $\gamma_{KPS\text{-}LT}$ (blue in Fig. 7) using our proposed CKD-LT or KPS-LT method may serve as a lower limit of $\gamma$ (see Sect. 2.3 for explanation), thus defining the uncertainty range of $\gamma$, as shown in Fig. 7.

To have a general cognition of the quantified divergence among the different types of uptake coefficients, we further present Figs. 8 and 9 as examinations of two specific experimental configurations. Each figure has two panels: with panel A showing the uptake coefficient corrected by the CKD and CKD-LT methods and panel B by the KPS and KPS-LT methods. The derivation of $\gamma_{eff}$ is based on Eqn. (C1). For Fig. 8, the experimental configuration of the soil coating case (Li et al., 2016) in Fig. 6 (solid circle) is used as input parameters for the diffusion correction. While an assumed configuration with higher volumetric flow rate $F$ and larger relative roughness height $\delta_r/R_0$ (see caption for details) are adopted for Fig. 9. As shown in both figures, the uncertainty range of $\gamma$ can be constrained by $\gamma_{CKD}$ and $\gamma_{CKD\text{-}LT}$, or $\gamma_{KPS}$ and $\gamma_{KPS\text{-}LT}$. In general, larger divergence, which corresponds to larger molecular diffusion or/and local turbulence effects, can be found at higher uptake coefficient magnitudes. The experimental configuration used for Fig. 8 results in a smaller difference of $\gamma_{CKD}$ against $\gamma_{CKD\text{-}LT}$ and $\gamma_{KPS}$ against $\gamma_{KPS\text{-}LT}$ than that for Fig. 9. This indicates that, for experiment design with rough coating, higher volumetric flow rate or/and larger relative roughness height will make the coating surface roughness effects more prominent. The higher values of the uptake coefficient derived using CKD and CKD-LT than those using KPS and KPS-LT respectively, can be due to the different algorithms employed for CKD and KPS (see Appendix C). At last, it should be noted that the whole discussion about surface roughness and the way the different diffusion correction methods are applied, are linked to the assumption that first-order reaction kinetics are granted, as mentioned upfront.

## 4. Conclusions

In this study, a new criterion is proposed to eliminate/minimize the potential effects of coating surface roughness on laminar flow in coated-wall flow tube experiments. Employment of this criterion in future flow tube experiments design can validate the application of conventional diffusion correction methods for uptake coefficient calculations. While keeping a coating film thickness well within the critical height $\delta_c$ to exclude potential surface roughness effects, flexible coated-wall flow tube design can also be achieved. For example, one can increase $\delta_c$ by adjusting flow tube geometric parameters (i.e., tube diameter and tube length) or flow velocity $V_{avg}$ to ensure not only an unaffected laminar flow pattern but also a situation-suitable residence time in flow tube reactors. We illustrate the application of this new criterion for previous investigations, and demonstrate its effectiveness in optimizing flow tube design and consolidating kinetic experimental results. Moreover, based on the CKD/KPS

methods, their modified versions (CKD-LT/KPS-LT) are proposed. The combinations of CKD/KPS and their modified versions can be used to quantify the maximum error of the calculated uptake coefficient ($\gamma_{CKD}$ or $\gamma_{KPS}$) when roughness-induced local turbulence occurs. And the real uptake coefficient $\gamma$ can be finally constrained by $\gamma_{CKD}$ and $\gamma_{CKD-LT}$ (or $\gamma_{KPS}$ and $\gamma_{KPS-LT}$).

## 5. Data availability

The Matlab code for CKD and CKD-LT is provided in Appendix E. The underlying research data can be accessed upon contact with Yafang Cheng (yafang.cheng@mpic.de), Hang Su (h.su@mpic.de) or Guo Li (guo.li@mpic.de).

## Appendix B

### Wall-roughness-induced error of $\gamma_{CKD}$ in LT regime: for previous flow tube studies

Local turbulence caused by rough surface coatings may introduce errors in the uptake coefficient derived from the
10 Brown/CKD/KPS methods (e.g., calculated uptake coefficient $\gamma_{CKD}$ or $\gamma_{KPS}$ illustrated in Fig. 7). We show here an example illuminating how this error estimation can be accomplished, by means of simulation under the pre-defined experimental configurations.

Figure A1 shows the maximum errors of $\gamma_{CKD}$ as a function of varying $\gamma_{eff}$ (A) and $\gamma_{CKD}$ (B). There, three different cases of $\delta_g/R_0$
are presented with all the other experimental configurations kept the same (see figure caption). For higher $\delta_g/R_0$, the errors of $\gamma_{CKD}$ are also larger, indicating that a thick and rough coating will generate more local turbulence and therefore have larger effects on derived uptake coefficients using the conventional molecular diffusion correction methods. Meanwhile, the errors are also closely related to the magnitude of $\gamma_{CKD}$ and $\gamma_{eff}$: when they are smaller than $10^{-4}$ the errors are inconspicuous, but beyond $10^{-4}$ the errors are apparent and considerably increase. The sharp increase of the error in Fig. A1 (A) is due to the fact that there is a
region where $\gamma_{CKD}$ is very sensitive to variations of the measured penetration $C/C_0$ as $\gamma_{CKD}$ getting close to 1 (i.e., the non-ideal region in Fig. A2). Compared to molecular diffusion, the roughness-induced turbulent transport may result in a lower $C/C_0$ which corresponds to a significant error of $\gamma_{CKD}$. In previous flow tube studies where local turbulence could not be avoided (LT regime), Fig. A1 can be used to estimate the potential maximum errors of the calculated $\gamma_{CKD}$. In order to guide flow tube designers to estimate the potential errors of their derived high uptake coefficient using our method, a tutorial derivation procedure for
$\gamma_{CKD}/\gamma_{CKD-LT}$ versus $\gamma_{CKD}$ or $\gamma_{eff}$ is further presented in Appendix D.

## Appendix C

### Comparison between KPS and CKD

The KPS method is a recently developed analytical approximation method. The derivation of KPS is based on kinetic flux model framework and models describing interactions of gas species with aerosols in combination with the diffusion limitation theory
for gas and particle uptake on a tube wall (Knopf et al., 2015, and references therein). This approximation method circumvents the complex operation procedures of previous numerical methods (e.g., the Brown and CKD methods), and therefore can be applied in a simpler way. As analyzed in KPS, the effective uptake coefficient $\gamma_{eff}$ can be experimentally determined as (Knopf et al., 2015):

$$\gamma_{eff} = \frac{d}{\omega \times t} \ln\left(\frac{C_0}{C}\right)$$

(C1)

where $d$ is flow tube diameter, $\omega$ is mean molecular speed of the gas reactant, $t$ is residence time of the gas reactant within the coated-wall region, $C_0$ and $C$ are gas reactant concentration at the flow tube inlet and outlet, respectively. After correction for gas molecular diffusion effects, the real uptake coefficient $\gamma$ is derived as follows:

$$\gamma = \frac{\gamma_{eff}}{1 - \gamma_{eff} \dfrac{3}{2 N_{Shw}^{eff} \times Kn}}$$

(C2)

in which $N_{Shw}^{eff}$ is the effective Sherwood number and $Kn$ is the Knudsen number, which can be expressed respectively as:

$$N_{Shw}^{eff} = 3.6568 + \frac{0.0978}{z^* + 0.0154} \quad \text{with} \quad z^* = L \times (\frac{\pi}{2}) \times \left(\frac{D}{F}\right)$$

(C3)

$$Kn = \frac{2\lambda}{d} \quad \text{with} \quad \lambda = \frac{3D}{\omega}$$

(C4)

where $z^*$ is dimensionless axial distance, $L$ is length of the coated-wall region, $D$ is molecular diffusion coefficient of the gas reactant within the carrier gas at experimental conditions, $F$ is volumetric flow rate of the fluid and $\lambda$ is mean free path of the gas reactant.

The CKD method in the present study is based on directly solving the differential equation, which is provided by Murphy and Fahey (1987) and used for description of the gas reactant concentration as a function of axial and radial position in a flow tube. Thus this CKD method can possess higher accuracy than the previously used CKD interpolation method or the KPS method (Knopf et al., 2015;Li et al., 2016).

As shown in Fig. A2, with ideal laminar flow (i.e., without any local turbulence, LF regime) the KPS and CKD show perfect agreement for the derived uptake coefficient in the fractional loss range of 0.452 to 1 (shaded area in panel A). Due to the different algorithms employed, however, the CKD method (Murphy and Fahey, 1987;Cooney et al., 1974;Davis, 1973;Li et al., 2016) and the KPS method (Knopf et al., 2015) could derive contrasting uptake coefficient values when local turbulence occurs. If a fractional loss is larger than the critical fractional loss value (i.e., $1 - C/C_0 > 0.452$, in panel B), e.g., because of enhanced mass transport towards the coated-wall due to local turbulence, the KPS results in a negative uptake coefficient (blue dashed line in Fig. A2) while the CKD has no solution. From Eqn (C1), it can be found that an unrealistically high fractional loss can lead to a high $\gamma_{eff}$, which may cause a negative denominator in Eqn (C2) and therefore a derived negative uptake coefficient. For a fractional loss value smaller than 0, both methods derive negative uptake coefficients implying emissions of gas reactants from the coating (i.e., $C/C_0 > 1$, in panel B).

**Appendix D**

**Derivation procedure of $\gamma_{CKD}/\gamma_{CKD-LT}$ versus $\gamma_{CKD}$ or $\gamma_{eff}$**

Derivation of $\gamma_{CKD}/\gamma_{CKD-LT}$ versus $\gamma_{CKD-LT}$ or $\gamma_{eff}$ is based on a combination of the modified CKD method (CKD-LT) and the CKD method (a CKD-based method using Matlab) which was described in our previous study (Li et al., 2016). The derivation principle is shown in Fig. A3. For one specific experiment configuration, both CKD and CKD-LT can generate a correlation

table (i.e., $Table_{CKD}$ for CKD and $Table_{CKD-LT}$ for CKD-LT) with its first column being penetration (i.e., $C_{CKD}/C_0$ or $C_{CKD-LT}/C_0$) and the second column the corresponding uptake coefficient ($\gamma_{CKD,n}$ or $\gamma_{CKD-LT,n}$), and their one-to-one correspondence is indicated by the same subscripts (e.g., $j$, $k$, etc.), as shown in Fig. A3. The abbreviations and symbols are explained in Appendix A. With local turbulence, a penetration ($C/C_0$) obtained from flow tube experiments corresponds to one specific uptake coefficient: in $Table_{CKD}$ this uptake coefficient is the calculated uptake coefficient $\gamma_{CKD}$ and in $Table_{CKD-LT}$ it refers to the uptake coefficient $\gamma_{CKD-LT}$. That is, with one identified $C/C_0$ the corresponding $\gamma_{CKD}$ and $\gamma_{CKD-LT}$ can be derived using CKD and CKD-LT respectively, and $\gamma_{CKD}/\gamma_{CKD-LT}$ can thereafter be determined.

In order to facilitate flow tube designers to evaluate $\gamma_{CKD}/\gamma_{CKD-LT}$ basing on their own experiment configurations, a tutorial derivation procedure is shown as following, and the $\gamma_{CKD}/\gamma_{CKD-LT}$ versus $\gamma_{CKD}$ derivation details of the case (the solid circle in Fig. 6) studied in the work by Li et al. (2016) are further elucidated as a derivation example.

1.  Input experimental parameters into CKD and CKD-LT models:

    For CKD and CKD-LT model calculation, the input parameters include: coated-wall region length $L$, volume flow rate $F$, flow tube radius $R_0$, the ratio of geometric coating thickness to tube radius $\delta_g/R_0$, the ratio of coating roughness height to geometric coating thickness $\delta_r/\delta_g$, experimental temperature $T$, experimental pressure $P$, mean molecular speed of the gas reactant $\omega$, diffusion coefficient of the gas reactant $D$.

    EXAMPLE: $L = 0.25$ m, $F = 1\times10^{-3}/60$ m$^3$/s, $R_0 = 0.0035$ m, $\delta_g/R_0 = 0.15$, $\delta_r/\delta_g = 0.2$, $T = 296$ K, $P = 101$ kPa, $\omega = 457.16$ m/s (gas reactant is HCHO), $D = 1.77\times10^{-5}$ m$^2$/s (HCHO diffusion within nitrogen at 296 K and 101 kPa).

2.  Models output penetration versus uptake coefficient results:

    CKD: the model calculation results are saved as an Excel file (i.e., $Table_{CKD}$ in Fig. A.3), with its first column as the penetration $C/C_0$ (i.e., $C_{CKD}/C_0$) and the second column as the calculated uptake coefficient $\gamma_{CKD}$ (i.e., $\gamma_{CKD,n}$).

    CKD-LT: the model calculation results are saved as an Excel file (i.e., $Table_{CKD-LT}$ in Fig. A.3), with its first column as the penetration $C/C_0$ (i.e., $C_{CKD-LT}/C_0$) and the second column as the uptake coefficient $\gamma_{CKD-LT}$ (i.e., $\gamma_{CKD-LT,n}$).

3.  Derive $\gamma_{CKD}/\gamma_{CKD-LT}$ versus $\gamma_{CKD}$ or $\gamma_{eff}$:

    For previous flow tube experiments which might be influenced by coating surface roughness, a measured penetration $C/C_0$ can point to a corresponded calculated uptake coefficient $\gamma_{CKD}$ using the CKD model generated table ($Table_{CKD}$). Meanwhile, this measured $C/C_0$ can also match an uptake coefficient $\gamma_{CKD-LT}$ using the CKD-LT model generated table ($Table_{CKD-LT}$). Then $\gamma_{CKD}/\gamma_{CKD-LT}$ versus $\gamma_{CKD}$ can be derived. On the other hand, the identified $C/C_0$ can be used for Eqn (C1) to derive $\gamma_{eff}$, and $\gamma_{CKD}/\gamma_{CKD-L}$ versus $\gamma_{eff}$ can be derived.

    EXAMPLE: $C/C_0 = 0.34$, $\gamma_{CKD} = 5.50\times10^{-5}$, $\gamma_{CKD-LT} = 5.29\times10^{-5}$, $\gamma_{eff} = 4.83\times10^{-5}$, $\gamma_{CKD}/\gamma_{CKD-LT} = 1.04$.

**Appendix E**

**Matlab code for CKD and CKD-LT**

**1. CKD**

```
% Basic Information
%-----------------------------------------------------------------------
```

```matlab
% Model Name: CKD
% Model Description: Derive uptake coefficient under merely molecular
%                    diffusion conditions (molecular diffusion correction)
% Developed by: Guo Li, Yafang Cheng, Hang Su and Ulrich Pöschl
% Contact: guo.li@mpic.de
% Developed at: 25.October.2017
% References: Murphy, D. M. and Fahey, D. W., Analytical Chemistry, 1987
%             Li,G.,et al., Atmos.Chem.Phys.,2016;
%-------------------------------------------------------------------------

% How to Use
%-------------------------------------------------------------------------
% 1st step: Input parameters according to the experimental configuration
% 2nd step: Save and run the Main function
% 3rd step: After running the function, check the output Excel in the folder
%           where the code is located
%-------------------------------------------------------------------------

function Main
% Main function
% Input Parameters
%*************************************************************************
% The length of coated-wall flow tube L, m
L = 0.25;
% The sample volume flow rate F, m^3/s
F = 1*10^(-3)/60;
% Temperature at standard conditions T0, K
T0 = 273;
% Pressure at standard conditions P0, kPa
P0 = 101;
% Temperature at experimental conditions T, K
T = 296;
% Pressure at experimental conditions P, kPa
P = 101;
% The minimum value of the uptake coefficient g, g_min
g_min = 1e-7;
% The maximum value of the uptake coefficient g, g_max
g_max = 1e-4;
% The number of g between g_min and g_max, g_n
g_n = 1000;
% Mean molecular velocity of the gas analyte v, m/s
global v
v = 457.16;
% The ratio between geometric coating thickness δg and tube radius R0, a
global a
a = 0.15;
% The ratio between roughness height δr and geometric coating thickness δg, b
global b
b = 0.2;
% Flow tube radius without coating R0, m
global R0
R0 = 0.0035;
% The diffusion coefficient of gas analyte at T and P, D, m^2/s
global D
D = 0.0000177;
%*************************************************************************
% Input END
t0=L*pi*D/(2*F)*(T0/T)*(P/P0);
Pdex1(t0,g_min,g_max,g_n)
%-------------------------------------------------------------------------
```

```matlab
function N = N_f(g)
% Sherwood Number
global R0
global a
global b
global v
global D
R = R0*(1-a+0.5*b*a);
N = 0.5*(v*R/D).*g./(2-g);
%-------------------------------------------------------------------------
function u0 = Pdex1ic(x)
% Initial conditions
u0 = 1;
%-------------------------------------------------------------------------
function [pl,ql,pr,qr] = Pdex1bc(xl,ul,xr,u,t)
% Boundary conditions
global g_i;
pl = 0;
ql = 0;
pr = N_f(g_i)*u;
qr = 1;
%-------------------------------------------------------------------------
function [c,f,s] = Pdex1pde(x,t,u,DuDx)
% Partial differential equation setting
c = 1-x^2;
f = DuDx;
s = 0;
%-------------------------------------------------------------------------
function Pdex1(t0,g_min,g_max,g_n)
% Partial differential equation
global g_i
global a
m = 1;
x = linspace(0,1,100);
t = linspace(0,t0,100);
g = linspace(g_min,g_max,g_n);
h = waitbar(0,'Please wait...');
steps = length(g);
for i=1:length(g)
    g_i = g(i);
    sol = pdepe(m,@Pdex1pde,@Pdex1ic,@Pdex1bc,x,t);
    u = sol(:,:,1);
    N_f(g(i))
    end_mean_u(i) = mean(u(end,:));
    waitbar(i / steps)
end
    A = [end_mean_u',g'];
close(h)
table_g = [end_mean_u',g'];

% Output Results
%-------------------------------------------------------------------------
xlswrite(['results',num2str(a),num2str(g_min),'.xls'], table_g);
%-------------------------------------------------------------------------
```

**2. CKD-LT**

% Basic Information

```matlab
%-------------------------------------------------------------------------
% Model Name: CKD-LT
% Model Description: Derive uptake coefficient under molecular diffusion
%                    and surface-roughness-induced local turbulence
%                    conditions (for local turbulence effects estimation
%                    when combined with the CKD model)
% Developed by: Guo Li, Yafang Cheng, Hang Su and Ulrich Pöschl
% Contact: guo.li@mpic.de
% Developed at: 25.October.2017
% References: Murphy, D. M. and Fahey, D. W., Analytical Chemistry, 1987;
%             Li,G.,et al., Atmos.Chem.Phys.,2016;
%-------------------------------------------------------------------------

% How to Use
%-------------------------------------------------------------------------
% 1st step: Input parameters according to the experimental configuration
% 2nd step: Save and run the Main function
% 3rd step: After running the function, check the output Excel in the folder
%           where the code is located
%-------------------------------------------------------------------------

function Main
% Main function
% Input Parameters
%*************************************************************************
% The length of the coated-wall flow tube L, m
L = 0.25;
% The minimum value of the uptake coefficient g, g_min
g_min = 1e-7;
% The maximum value of the uptake coefficient g, g_max
g_max = 1e-4;
% The number of g between g_min and g_max, g_n
g_n = 1000;
% The sample volume flow rate F, m^3/s
global F
F = 1*10^(-3)/60;
% Temperature at standard conditions T0, K
global T0
T0 = 273;
% Pressure at standard conditions P0, kPa
global P0
P0 = 101;
% Temperature at experimental conditions T, K
global T
T = 296;
% Pressure at experimental conditions P, kPa
global P
P = 101;
% Mean molecular velocity of the gas analyte at T and P, v, m/s
global v
v = 457.16;
% The ratio between geometric coating thickness δg and tube radius R0, a
global a
a = 0.15;
% The ratio between roughness height δr and geometric coating thickness δg, b
global b
b = 0.2;
% Flow tube radius without coating R0, m
global R0
R0 = 0.0035;
```

```matlab
    % Diffusion coefficient of the gas analyte at T and P, D, m^2/s
    global D
    D = 0.0000177;
    %*************************************************************************
    % Input END
    F1 = F*(1-a)^2/(1-a+0.5*a*b)^2;
    t0 = L*pi*D/(2*F1)*(T0/T)*(P/P0);
    Pdex1(t0,g_min,g_max,g_n)
    %-------------------------------------------------------------------------
    function N = N_f(g)
    % Sherwood Number
    global R0
    global a
    global v
    global D
    R = R0*(1-a);
    N = 0.5*(v*R/D).*g./(2-g);
    %-------------------------------------------------------------------------
    function u0 = Pdex1ic(x)
    % Initial conditions
    u0 = 1;
    %-------------------------------------------------------------------------
    function [pl,ql,pr,qr] = Pdex1bc(xl,ul,xr,u,t)
    % Boundary conditions
    global g_i;
    pl = 0;
    ql = 0;
    pr = N_f(g_i)*u;
    qr = 1;
    %-------------------------------------------------------------------------
    function [c,f,s] = Pdex1pde(x,t,u,DuDx)
    % Partial differential equation setting
    c = 1-x^2;
    f = DuDx;
    s = 0;
    %-------------------------------------------------------------------------
    function Pdex1(t0,g_min,g_max,g_n)
    % Partial differential equation
    global g_i
    global a
    m = 1;
    x = linspace(0,1,100);
    t = linspace(0,t0,100);
    g = linspace(g_min,g_max,g_n);
    h = waitbar(0,'Please wait...');
    steps = length(g);
    for i=1:length(g)
        g_i = g(i);
        sol = pdepe(m,@Pdex1pde,@Pdex1ic,@Pdex1bc,x,t);
        u = sol(:,:,1);
        N_f(g(i))
        end_mean_u(i)= mean(u(end,:));
        waitbar(i / steps)
    end
    close(h)
    table_g = [end_mean_u',g'];

    % Output Results
    %-------------------------------------------------------------------------
    xlswrite(['results',num2str(a),num2str(g_min),'.xls'], table_g);
```

**Acknowledgments**

This study was supported by the Max Planck Society (MPG) and National Natural Science Foundation of China (Grant No. 41330635 and 91644218). Guo Li acknowledges the financial support from the China Scholarship Council (CSC). Y.C. and H.S. conceived the study. G.L., Y.C., H.S. and U.P. developed the methods. G.L. performed data analysis. Y.C., H.S., U.P., U.K., M.A. and M.S. discussed the results. G.L., Y.C. and H.S. wrote the manuscript with inputs from all co-authors.

**List of Tables:**

**Appendix A**

List of abbreviations and symbols

| Abbreviation/Symbol | Meaning |
| --- | --- |
| CKD | Cooney-Kim-Davis method for molecular diffusion correction (numerical solution) |
| CKD-LT | a modified CKD method to account for roughness-induced local turbulence effects |
| KPS | Knopf-Pöschl-Shiraiwa method for molecular diffusion correction (analytical approximation) |
| KPS-LT | a modified KPS method to account for roughness-induced local turbulence effects |
| $\gamma$ | real uptake coefficient |
| $\gamma_{CKD}$ | uptake coefficient derived using the CKD method |
| $\gamma_{CKD\text{-}LT}$ | uptake coefficient derived using the CKD-LT method |
| $\gamma_{KPS}$ | uptake coefficient derived using the KPS method |
| $\gamma_{KPS\text{-}LT}$ | uptake coefficient derived using the KPS-LT method |
| $\gamma_{eff}$ | experimentally determined effective uptake coefficient |
| $Re$ | Reynolds number |
| $\rho$ | density of the fluid passing through the flow tube |
| $F$ | volumetric flow rate |
| $V_{avg}$ | average velocity of the fluid (i.e., the volumetric flow rate divided by the cross sectional area of the flow tube) |
| $d$ | inner diameter of the coated-wall flow tube |
| $\mu$ | dynamic viscosity of the fluid |
| $v$ | kinematic viscosity of the fluid |
| $\delta_r$ | roughness height |
| $\delta_l$ | thickness of the laminar boundary layer |
| $\delta_c$ | critical height calculated using the Eqn. (2) |
| $\delta_g$ | geometric coating thickness |
| $\delta_m$ | mass-based coating thickness |
| $L_c$ | characteristic length |
| $L$ | coated-wall region length |
| $R_0$ | flow tube radius without coating |
| $R_m$ | flow tube radius calculated using $\delta_m$, (i.e., $R_m = R_0 - \delta_m$) |
| $R_g$ | flow tube radius calculated using $\delta_g$, (i.e., $R_g = R_0 - \delta_g$) |
| LF regime | laminar flow regime shown in Fig. 3 (A) and Fig. 6 |
| LT regime | local turbulence regime shown in Fig. 3 (B) and Fig. 6 |
| $C$ | gas reactant concentration at the flow tube outlet |

| | |
|---|---|
| $C_0$ | gas reactant concentration at the flow tube inlet |
| $C/C_0$ | penetration |
| $1 - C/C_0$ | fractional loss |
| $\omega$ | mean molecular speed of the gas reactant |
| $t$ | interaction time between the gas reacant and the coated-wall (i.e., residence time) |
| $N_{Shw}^{eff}$ | effective Sherwood number |
| $Kn$ | Knudsen number |
| $z^*$ | dimensionless axial distance |
| $D$ | gas diffusion coefficient of the gas reactant |
| $\lambda$ | mean free path of the gas reactant |
| $C_{CKD}/C_0$ | penetration in the CKD generated table (Table$_{CKD}$) |
| $\gamma_{CKD, n}$ | uptake coefficient in the CKD generated table (Table$_{CKD}$) |
| $(C_{CKD}/C_0)_j$ | penetration at the $j$th row in table (Table$_{CKD}$) |
| $\gamma_{CKD, j}$ | uptake coefficient at the $j$th row in table (Table$_{CKD}$) |
| $C_{CKD\text{-}LT}/C_0$ | penetration in the CKD-LT generated table (Table$_{CKD\text{-}LT}$) |
| $\gamma_{CKD\text{-}LT, n}$ | uptake coefficient in the CKD-LT generated table (Table$_{CKD\text{-}LT}$) |
| $(C_{CKD\text{-}LT}/C_0)_k$ | penetration at the $k$th row in table (Table$_{CKD\text{-}LT}$) |
| $\gamma_{CKD\text{-}LT, k}$ | uptake coefficient at the $k$th row in table (Table$_{CKD\text{-}LT}$) |

**List of Figures:**

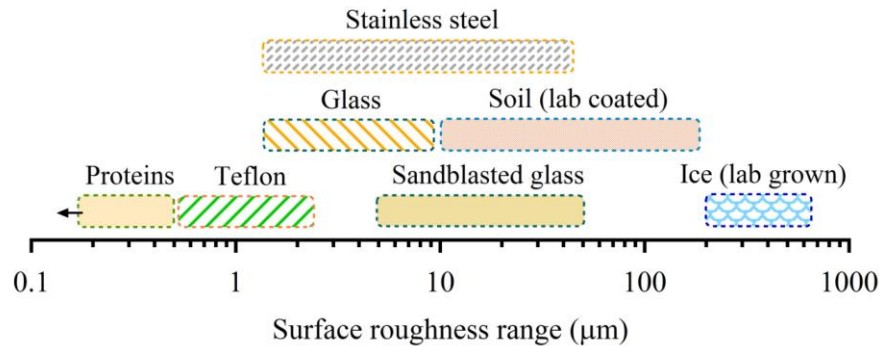

Figure 1. Typical surface roughness for materials commonly used in flow tube gas uptake and kinetic experiments. Data sources: https://neutrium.net/fluid_flow/absolute-roughness/ and http://www.edstech.com/design-tools.html. The soil roughness refers to Li et al. (2016) and the ice roughness refers to Onstott et al. (2013) and Landy et al. (2015).

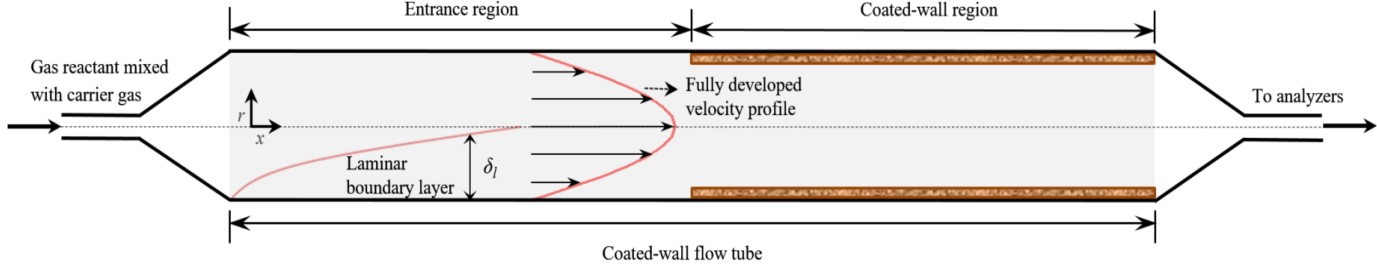

**Figure 2**. Development of laminar boundary layer and flow velocity profile within the coated-wall flow tube used for soil uptake experiments (d = 7 mm, L = 250 mm, Li et al. (2016)).

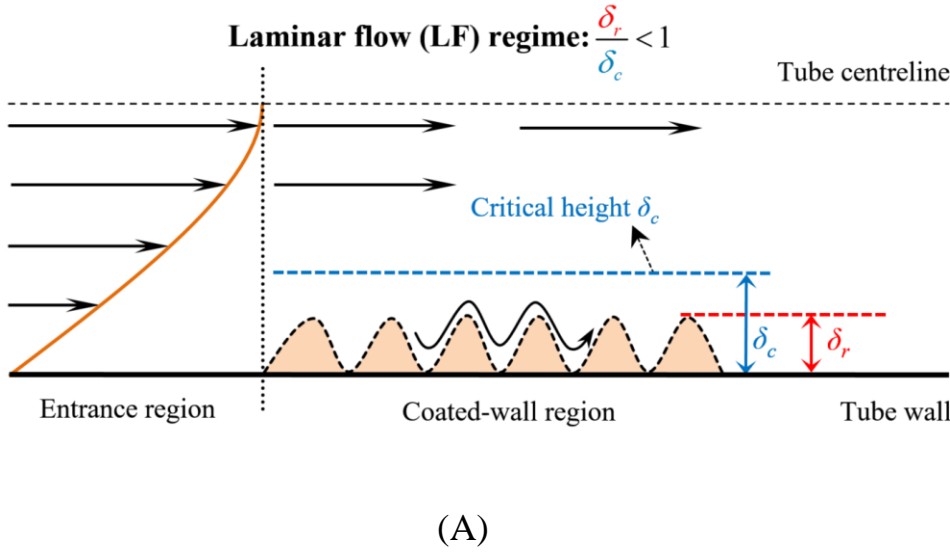

(A)

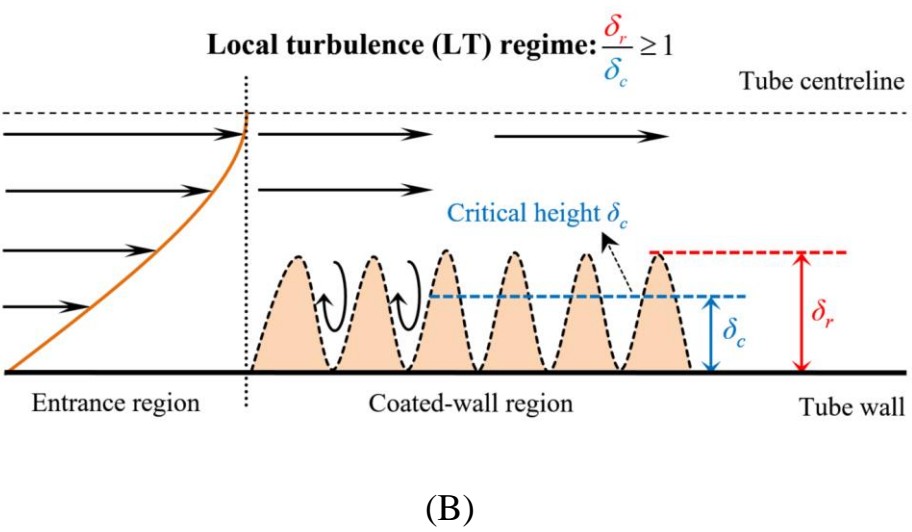

(B)

**Figure 3.** Schematic of the critical height $\delta_c$ and its related flow conditions in a flow tube with rough coatings. Upstream of the coated-wall region, the entrance region is designed to warrant well-developed laminar flow conditions. Two cases of tube coatings reflect different impacts of a roughness element with varying height $\delta_r$ on flow patterns.

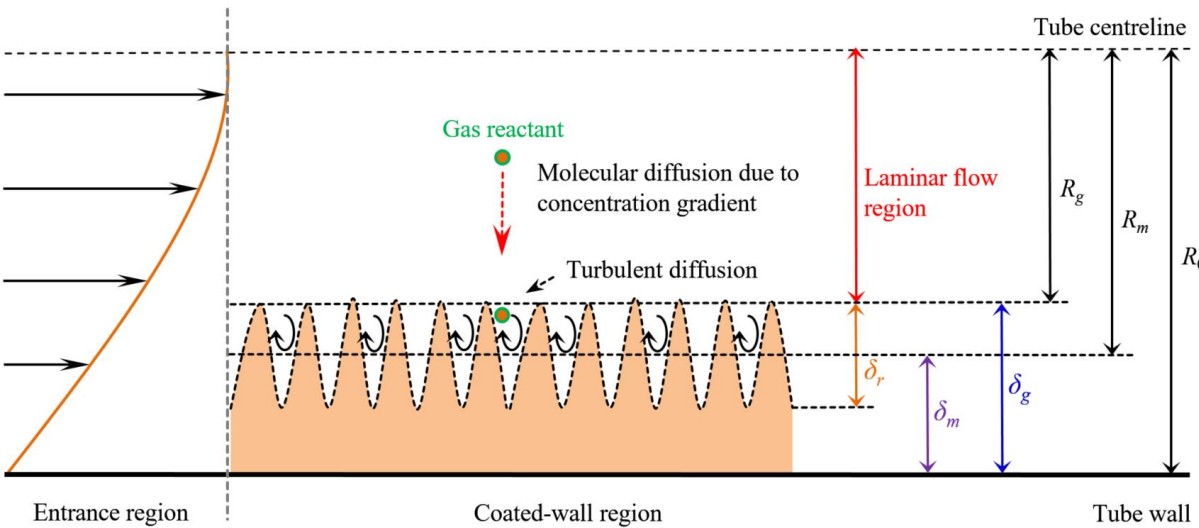

**Figure 4.** Illustration of the variables used for the CKD-LT and KPS-LT methods: $\delta_r$, roughness height; $\delta_m$, mass-based coating thickness; $\delta_g$, geometric coating thickness; $R_g$, calculated flow tube radius based on $\delta_g$; $R_m$, calculated flow tube radius based on $\delta_m$; $R_0$, flow tube radius without coating.

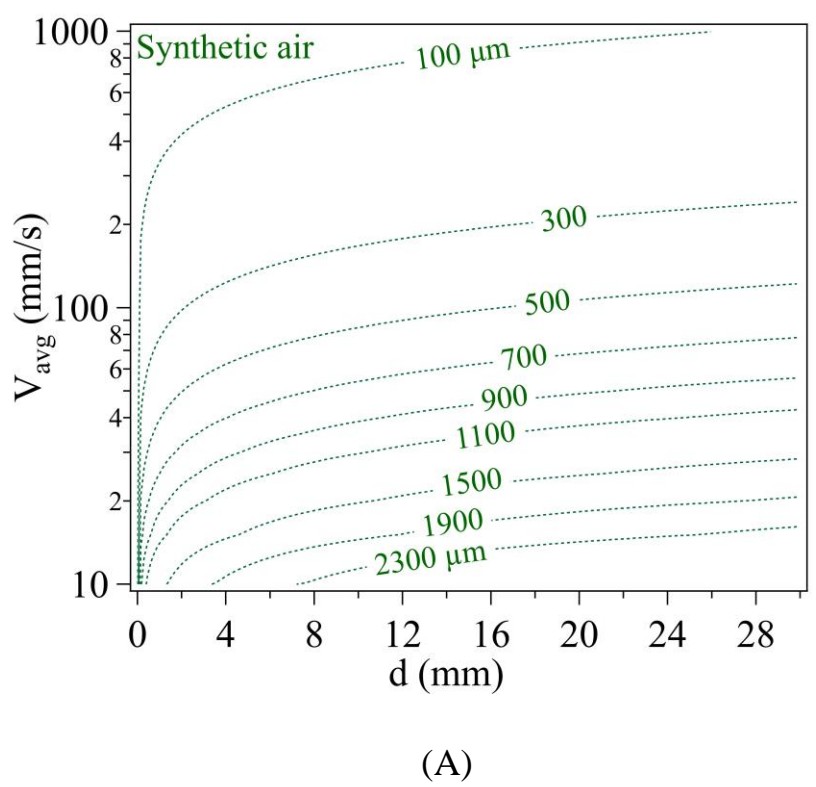

(A)

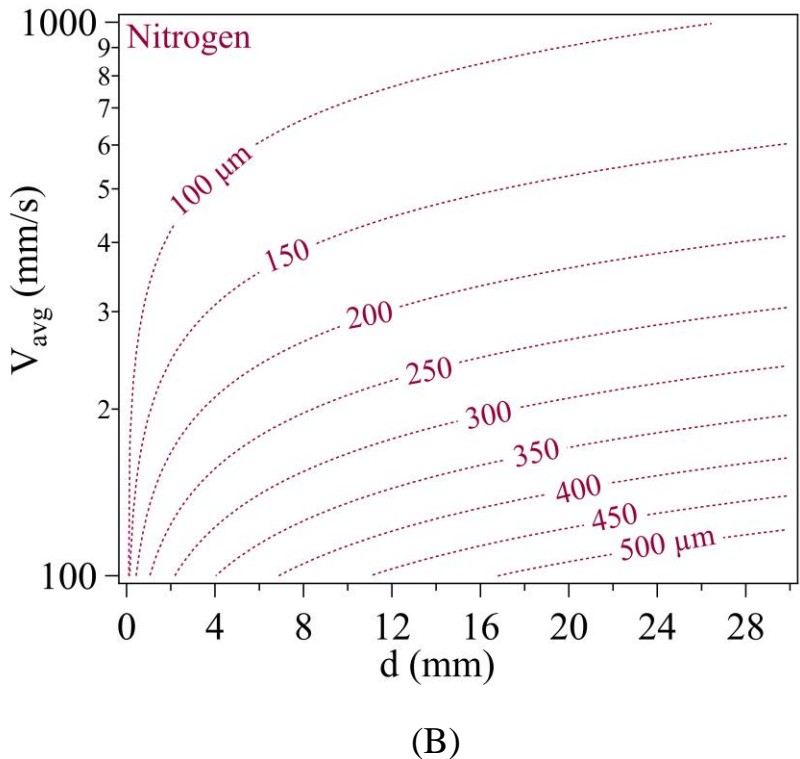

(B)

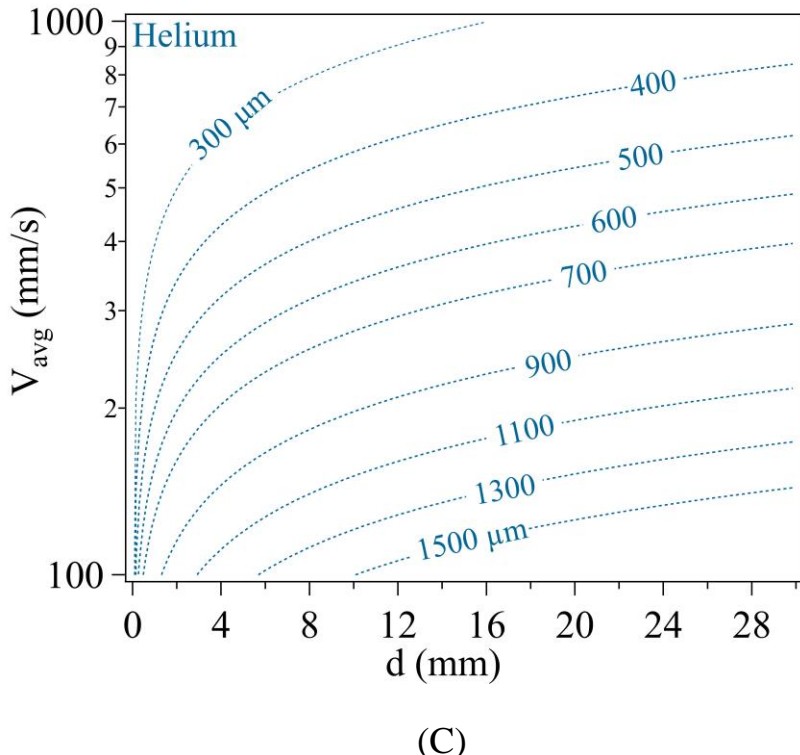

(C)

**Figure 5.** Calculated critical height $\delta_c$ (dash-dotted lines) versus varying tube diameter $d$ and flow velocity $V_{avg}$ in flow tube experiments with carrier gases of synthetic air (A), nitrogen (B) and helium (C), respectively.

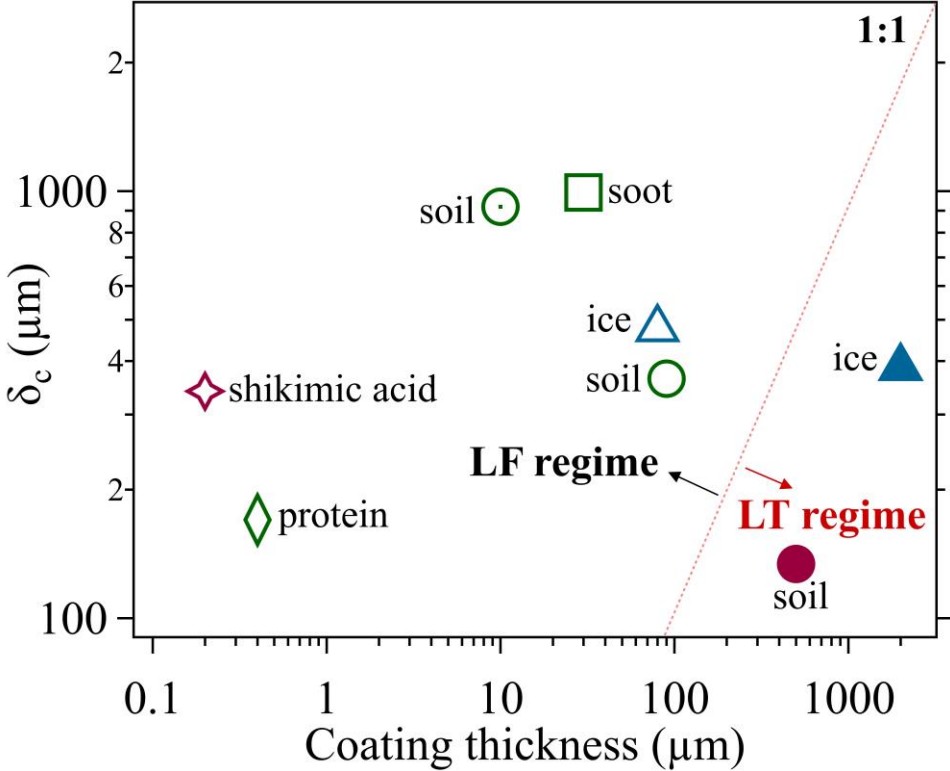

**Figure 6.** Representative coating thickness in previous coated-wall flow tube studies, versus the calculated critical height $\delta_c$ (based on their experimentally adopted $d$ and $V_{avg}$). The color of the symbols indicates the different types of carrier gases employed: synthetic air (green symbols), nitrogen (purple) and helium (blue). References for the coatings summarized here are: diamond (Shiraiwa et al., 2011), square (Monge et al., 2010), open circle (Donaldson et al., 2014a;Donaldson et al., 2014b),open circle with center (Wang et al., 2012), solid circle (Li et al., 2016), star  (Steimer et al., 2015), solid triangle (McNeill et al., 2006) and open triangle (Petitjean et al., 2009).  LF and LT refers to laminar flow and local turbulence, respectively.

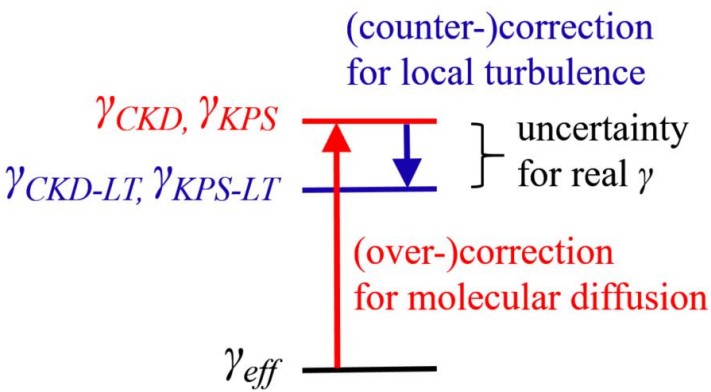

**Figure 7.** Schematic of different types of uptake coefficient and their divergences due to molecular diffusion and local turbulence effects. The uncertainty of $\gamma$ is constrained by $\gamma_{CKD}$ and $\gamma_{CKD-LT}$, or $\gamma_{KPS}$ and $\gamma_{KPS-LT}$. Note that the degree of the divergences among these types of uptake coefficient depends on their magnitude, i.e., for lower uptake coefficient values no corrections are needed (see Figs. 8 and 9). Similarly, $\gamma_{CKD}$ and $\gamma_{KPS}$, or $\gamma_{CKD-LT}$ and $\gamma_{KPS-LT}$ may differ from each other depending on their magnitude (see Figs. 8 and 9, and Appendix C). The abbreviations and symbols are explained in Appendix A.

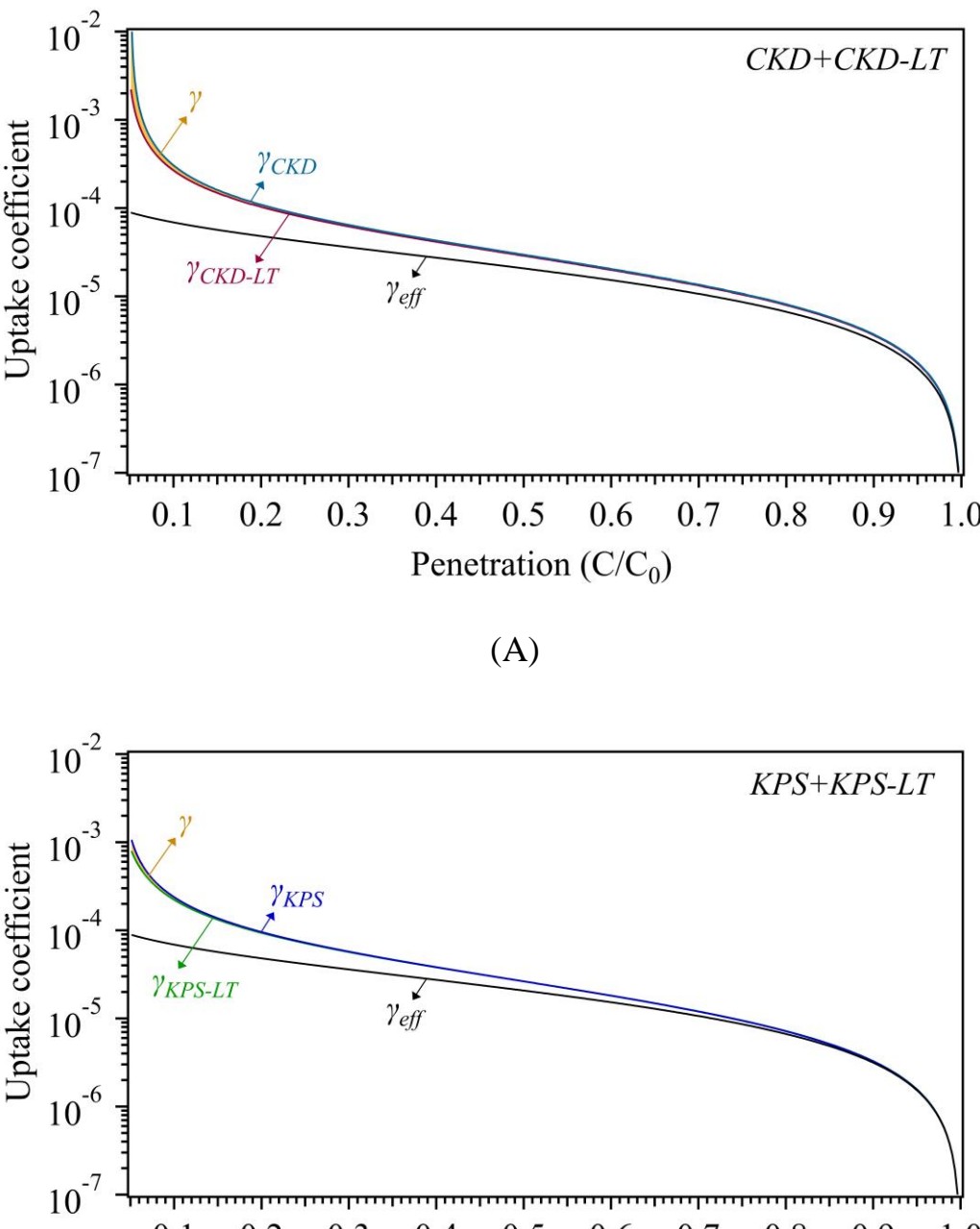

(A)

(B)

**Figure 8.** Schematic of different types of uptake coefficient versus the measured penetration ($C/C_0$), using both diffusion correction methods CKD (A) and KPS (B) as well as their modified versions i.e., CKD-LT and KPS-LT, to evaluate roughness-induced local turbulence effects. The yellow shaded area shows the uncertainty range of $\gamma$. Derivation of the uptake coefficient is based on the specific experimental parameters in our previous study (Li et al., 2016): gas reactant, HCHO; carrier gas, $N_2$; volumetric flow rate $F$ =1 L min$^{-1}$ at 1 atm and 296 K; flow tube dimension, d = 7 mm, L = 250 mm. The $\delta_g$ and $\delta_r$ of the soil coating are estimated using scanning electron microscopy: $\delta_g/R_0 = 0.15$, $\delta_r/\delta_g = 0.2$.

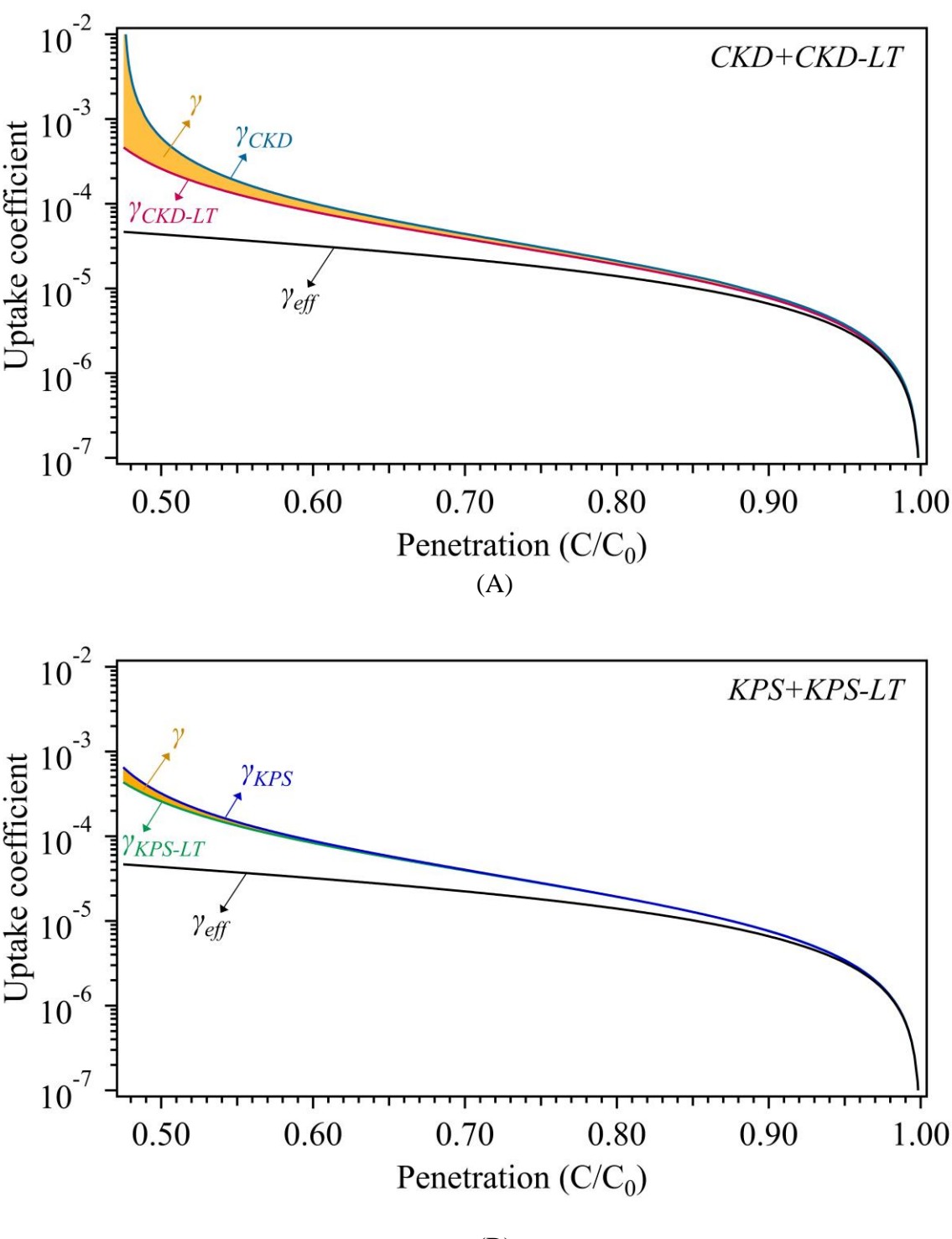

(A)

(B)

**Figure 9.** Schematic of different types of uptake coefficient versus the measured penetration ($C/C_0$), using both diffusion correction methods CKD (A) and KPS (B) as well as their modified versions i.e., CKD-LT and KPS-LT, to evaluate roughness-induced local turbulence effects. The yellow shaded area shows the uncertainty range of $\gamma$. Derivation of the uptake coefficient is based on the following assumptions: gas reactant, $O_3$; carrier gas, $N_2$; volumetric flow rate $F = 5$ L min$^{-1}$ at 1 atm and 298 K; flow tube dimension, d = 22 mm, L = 250 mm. $\delta_g$ and $\delta_r$ of the coating material are defined by $\delta_g/R_0 = 0.2$, $\delta_r/\delta_g = 0.5$. The choice of 0.5 for $\delta_r/\delta_g$ represents an extreme rough coating case.

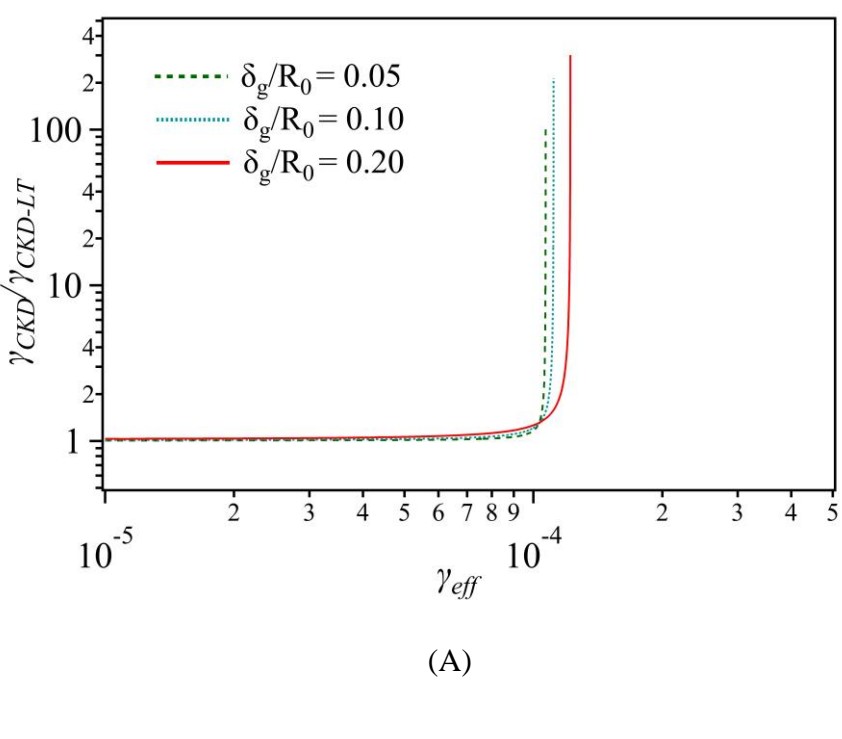

(A)

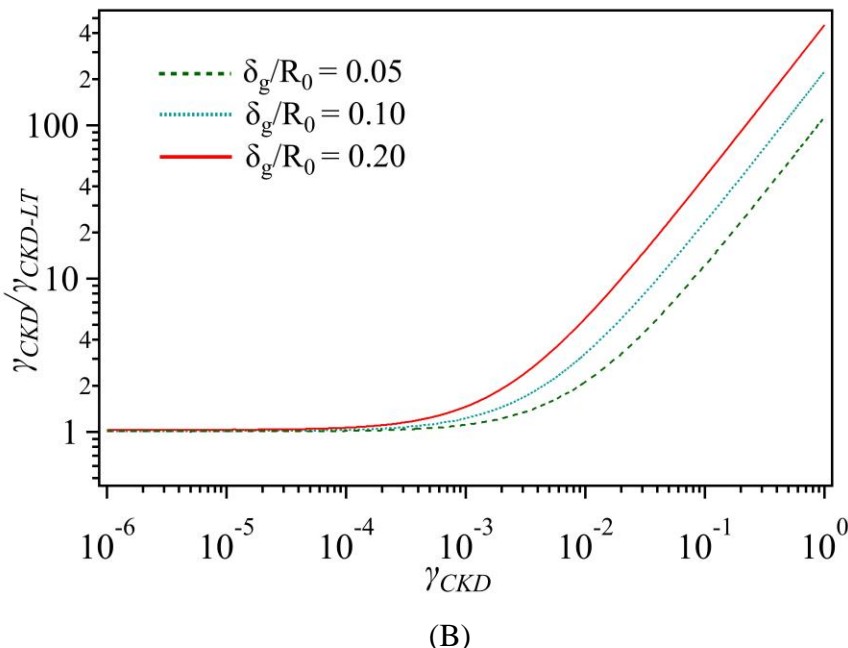

(B)

**Figure A1.** Maximum error of the CKD derived uptake coefficient ($\gamma_{CKD}$) relative to the CKD-LT derived uptake coefficient ($\gamma_{CKD-LT}$) versus changing $\gamma_{eff}$ (A) and $\gamma_{CKD}$ (B) for three cases with different ratio of the geometric coating thickness to tube radius ($\delta_g/R_0$). For derivation of this plot, the specific experimental configuration is: gas reactant, $O_3$; carrier gas, $N_2$; volumetric flow rate $F$ =1 L min$^{-1}$ at 1 atm and 298 K; flow tube dimension, d = 7 mm, L = 250 mm. The choices of $\delta_g/R_0$ cover the general ratio range in previous studies. The curves cannot be further extended due to reaching the limits of diffusion correction methods (see Appendix C).

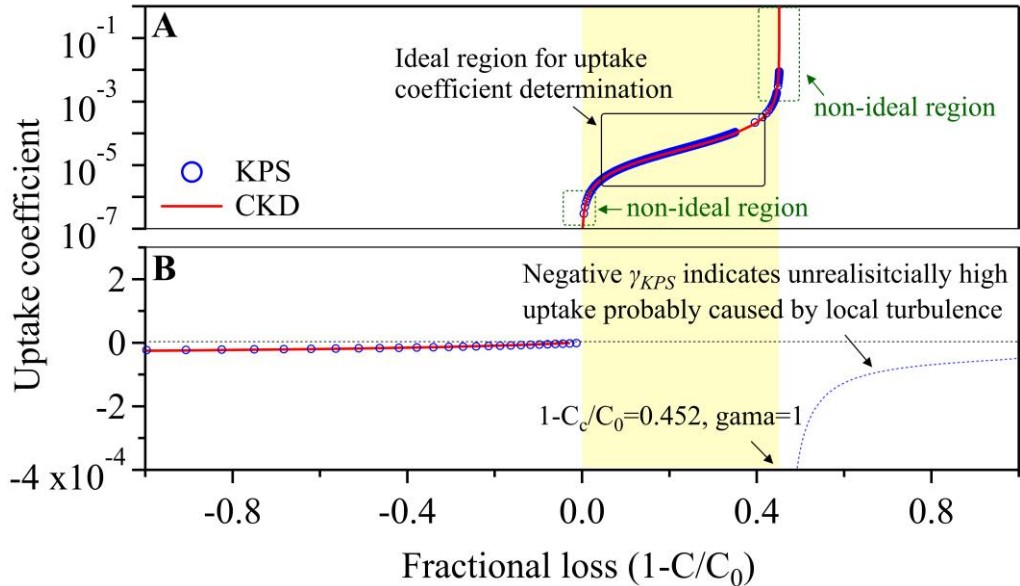

**Figure A2.** Comparisons between uptake coefficients (derived from KPS and CKD methods, respectively) versus the fractional loss. Panel (A) displays the derived positive uptake coefficients under laminar flow (LF) regime, and panel (B) the derived negative ones due to emission (the left) or local turbulence effect (the right). For derivation of this plot, the specific experimental configuration is: gas reactant, $SO_2$; carrier gas, synthetic air; volumetric flow rate $F = 4$ L min$^{-1}$ at 1 atm and 296 K; flow tube dimension, d = 17 mm, L = 200 mm.

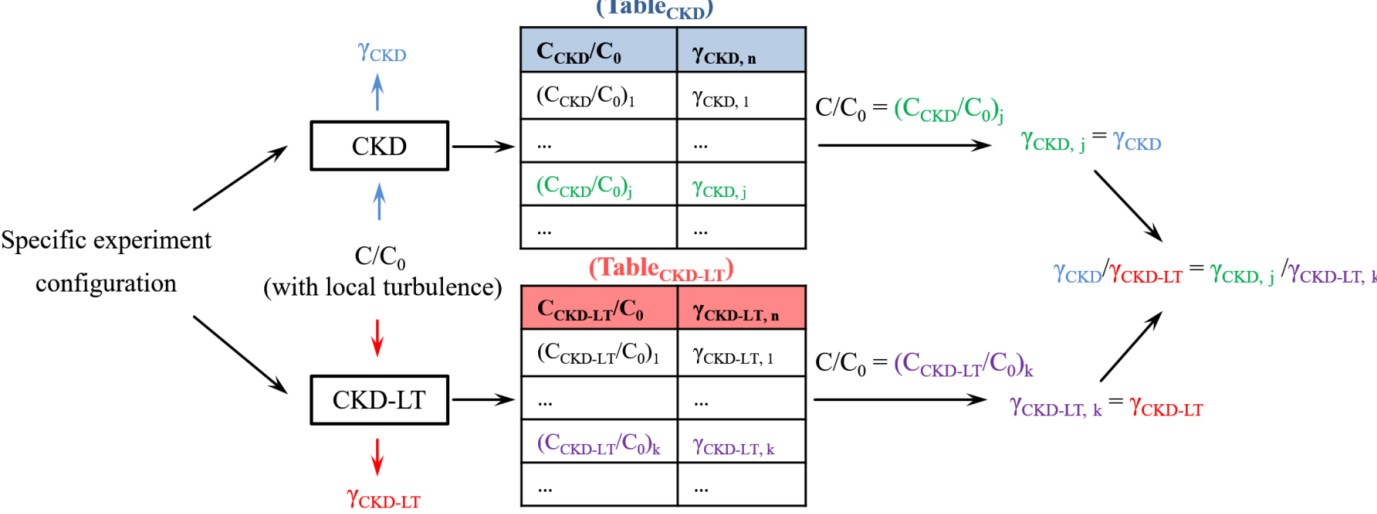

**Figure A3.** Schematic of the derivation principle for $\gamma_{CKD}/\gamma_{CKD\text{-}LT}$. The abbreviations and symbols are explained in Appendix A.

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
