# Peer review of "Technical Note: Influence of surface roughness and local turbulence on coated-wall flow tube experiments for gas uptake and kinetic studies"

_Atmospheric Chemistry and Physics, 2017_

## Referee Comment (RC1) · Anonymous Referee #1 · 17 May 2017

**Comments on "Technical Note: Influence of surface roughness and local turbulence on coated-wall flow tube experiments for gas uptake and kinetic studies"**

There has been a debate on whether the coating surface roughness could disturb the fully developed laminar flow in flow tube kinetic experiments and its effects were usually not well-quantified. This article give a new criterion to eliminate the potential effects of coating surface roughness on laminar flow in coated-wall flow tube experiments. They employ a critical height to provide an easy way of compromising different flow tube experimental parameters. The article also summarized previous flow tube investigations employing various coating materials and thicknesses and further evaluated by the proposed criterion of $\delta_c$, as an illustration of how this criterion can be applied. The article indicated that increasing $\delta_c$ by adjusting flow tube geometric parameters or $V_{avg}$ can reduce the effects of coating surface roughness on laminar flow in coated-wall flow tube experiments. The article is of interesting to readers, and is suitable for publication in this Journal. I recommended it to be accepted after minor revision.

Comments:

The most important of this article is to let the researchers to calculated more accuracy uptake coefficients by flow tube experiments. However, I feel that the discussion of Part 3.2 is not very enough. Please give more example on how to use the author's method to get the estimation of the potential error of the measured $\gamma_{eff}$ when $\gamma_{eff} > 10^{-3}$

---

## Referee Comment (RC2) · Anonymous Referee #2 · 26 May 2017

Overall Evaluation. Li et al. have conducted a detailed study of the effect of substrate roughness on the outcome of kinetics studies using coated-wall flow tubes. Uptake coefficients derived from coated wall flow tubes operated at or near atmospheric pressure must be corrected for radial concentration gradients within the tube, especially when uptake to the coated wall is highly efficient. Corrections are performed using the methods of Brown (1978), CDK (Murphy & Fahey, 1987), or KPS (Knopf et al. 2015) that are by now well established. However, no study to date has taken a systematical look at the effect of substrate surface roughness and resulting turbulent diffusion effects in the analysis of coated wall flow tube experiments. The authors outline the defining variables in defining surface roughness (e.g., roughness height, relative roughness, and the critical height) and describe how these parameters can be understood in the context of the widely used diffusion correction treatments. They then present a very useful method of identifying when uptake coefficients are negatively impacted by roughness-induced turbulence and provide a means to estimate the associated error in uptake coefficients due to this effect. For the most part, the approach is clear/simple to follow and the manuscript well written and presented.

Coated wall flow tubes are widely used for studying the kinetics of heterogeneous and multiphase chemistry reactions. Much of the uptake coefficient data found in the NASA or IUPAC evaluations that are used for modeling on the efficiency have been derived by this technique. When working with environmentally relevant substrates (e.g., ice, soil, mineral dust, etc.) the surface is inherently rough and practitioners (and reviewers of their manuscripts) have often speculated on the effects of surface roughness on the results. The authors demonstrate, using a handful of data extracted from the literature that such roughness effects must be considered; depending on the flow tube configuration, errors can be potentially significant. Given our dependence on accurately determined uptake coefficients in modeling heterogeneous processes, it is critical that we understand the effect of substrate roughness on the experiments used to measure them.

In my opinion, this study provides an important contribution to the literature since it will help the atmospheric community evaluate the quality of measured uptake coefficients and will provide experimentalists with a tool for designing a flow tube configuration that avoids errors due to turbulence created by surface roughness. I am highly supportive of publication. Below are a few minor comments the authors may wish to consider in revising the manuscript.

Specific Comments (listed by page: line#) 4:19: The critical height is introduced on this line, but it is only at the bottom of page 5 that we have a formal mathematical definition of the critical height. Between page 4 and 5 there is a discussion of figures that involves this critical height but an explicit definition was lacking. I would recommend

including a general/non mathematical definition of the critical height concept when it is first introduced; doing so would improve readability of section 2.1.

9:13: (Section A.2). I found this section somewhat tedious to read and not conducive to teaching the reader how to perform the calculations described. Perhaps this is due to the abbreviated format that relied on leading the reader through the flow charts in Figure A.3. I believe this section could be elaborated on in more of a tutorial fashion to increase reader comprehension and to allow the reader to more easily derive such plots themselves. In addition, the authors may wish to include a table of abbreviations in the supplementary files.

9:30: (Figure A.4). Two stacked graphs are included in figure A.4 and it took me a while to understand what they were referring to. My interpretation based on section A.3 on page 9 and the abbreviated figure caption is that the top panel refers to the condition where ideal laminar flow is considered for the various diffusion corrections, while the lower panel considers how the various correction methods break down when local turbulence in the laminar flow occurs. I recommend referring to the two panels as A and B in text and then including labels of the modeling conditions in either the graph or the associated figure caption.

---

## Author Comment (AC1) · 21 Jan 2018

**Response to Anonymous Referee #1**

We thank the reviewer for the constructive suggestions/comments. Below we provide a point-by-point response to individual comment (Reviewer comments and suggestions are in italics, responses and revisions are in plain font; revised parts in responses are marked with red color; page numbers refer to the modified ACPD version).

**Comments and suggestions:**

*The most important of this article is to let the researchers to calculated more accuracy uptake coefficients by flow tube experiments. However, I feel that the discussion of Part 3.2 is not very enough. Please give more example on how to use the author's method to get the estimation of the potential error of the measured $\gamma_{eff}$ when $\gamma_{eff} > 10^{-3}$.*

**Responses and Revisions:**

Good suggestion. We have revised the original text in Part 3.2 and moved it to Appendix B in the revised manuscript. In the current Part 3.2, we show a schematic of different types of uptake coefficients and their divergences due to molecular diffusion and local turbulence effects, and further two examples under different experimental configurations. This new version of Part 3.2 can help the readers gain an overall comprehension of different types of uptake coefficients and where their divergences originate from. With these as a basis, the readers can more easily understand our proposed methods to quantify local turbulence effect. Moreover, we have added a tutorial derivation procedure in Appendix D to provide a detailed guidance for the readers to use our proposed methods to estimate the potential errors of their measured high $\gamma_{eff}$ (e.g., $\gamma_{eff} > 10^{-3}$). The employed Matlab code for error evaluation have been added in Appendix E.

The revised original Part 3.2 is now shown in Appendix B:

[revised manuscript text omitted]

%----------------------------------------------------------------------"
```

To improve the readability and clarity of the whole manuscript and to help the readers easily understand and further utilize our proposed method to evaluate their flow tube experiment results and to make better flow tube coating design, additional revisions have been made as shown below.

Some symbols and abbreviations in this manuscript have been changed or re-defined:

$\varepsilon_{max}$ (coating thickness) $\longrightarrow$ $\delta_g$ (geometric coating thickness) and $\delta_m$ (mass-based coating thickness);

$\varepsilon$ (roughness height) $\longrightarrow$ $\delta_r$ (roughness height);

$\gamma_{eff}$ (calculated effective uptake coefficient) $\longrightarrow$ $\gamma_{CKD}$, $\gamma_{KPS}$ (the uptake coefficient derived using the conventional CKD/KPS methods, respectively);

M-CKD (a modified CKD method) $\longrightarrow$ CKD-LT (a modified CKD method to account for local turbulence);

$C/C_0$ (concentration transmittance) $\longrightarrow$ $C/C_0$ (penetration);

Case 1 $\longrightarrow$ LT regime (local turbulence regime);

Case 2 $\longrightarrow$ LF regime (laminar flow regime);

All the symbols/abbreviations can be found in Table A.1 in Appendix A.

The additional revisions are marked with red color and the whole revised manuscript is shown below.

[revised manuscript text omitted]

---

## Author Comment (AC2) · 21 Jan 2018

**Response to Anonymous Referee #2**

We thank the reviewer for the constructive suggestions/comments. Below we provide a point-by-point response to individual comment (Reviewer comments and suggestions are in italics, responses and revisions are in plain font; revised parts in responses are marked with red color; page numbers refer to the modified ACPD version).

**Comments and suggestions:**

*Specific Comments (listed by page: line#) 4:19: The critical height is introduced on this line, but it is only at the bottom of page 5 that we have a formal mathematical definition of the critical height. Between page 4 and 5 there is a discussion of figures that involves this critical height but an explicit definition was lacking. I would recommend including a general/non mathematical definition of the critical height concept when it is first introduced; doing so would improve readability of section 2.1.*

**Responses and Revisions:**

We thank the reviewer's helpful comments and recommendation. We have further provided a general and non-mathematical definition, in page 4, line 25, to improve readability of section 2.1:

"…In view of the special laminar boundary layer structure in flow tubes, we employ a critical height $\delta_c$, which defines the smallest scale within which local turbulence can occur (i.e., for scales smaller than $\delta_c$, local turbulence cannot exist, see Kolmogorov (1991)), to evaluate the influence of surface roughness on laminar flow patterns"

**Comments and suggestions:**

*9:13: (Section A.2). I found this section somewhat tedious to read and not conducive to teaching the reader how to perform the calculations described. Perhaps this is due to the abbreviated format that relied on leading the reader through the flow charts in Figure A.3. I believe this section could be elaborated on in more of a tutorial fashion to increase reader comprehension and to allow the reader to more easily derive such plots themselves. In addition, the authors may wish to include a table of abbreviations in the supplementary files.*

**Responses and Revisions:**

We thank the reviewer's comments and suggestions. In order to help the readers easily understand the original flow charts in Figure A.3, the flow charts have been simplified to serve as a pictorial description of the derivation principle of $\gamma_{CKD}/\gamma_{CKD-LT}$ versus $\gamma_{CKD-LT}$ or $\gamma_{CKD}$, and accordingly the illustrating texts have been modified. Meanwhile, a table of abbreviations and symbols used in the whole context has been included in the Appendix A. Moreover, to facilitate the readers to easily use our method to evaluate the potential errors of their measured high magnitudes of uptake coefficients, a detailed tutorial derivation procedure is added in Appendix D:

"**Appendix D**

[revised manuscript text omitted]

**Comments and suggestions:**

*9:30: (Figure A.4). Two stacked graphs are included in figure A.4 and it took me a while to understand what they were referring to. My interpretation based on section A.3 on page 9 and the abbreviated figure caption is that the top panel refers to the condition where ideal laminar flow is considered for the various diffusion corrections, while the lower panel considers how the various correction methods break down when local turbulence in the laminar flow occurs. I recommend referring to the two panels as A and B in text and then including labels of the modeling conditions in either the graph or the associated figure caption.*

**Responses and Revisions:**

We thank the reviewer's comments and suggestions. The original Figure A.4 has been further simplified (in the revised version shown as Figure A2), that is, only the data with diffusion corrections employing KPS and CKD respectively, are displayed. The data for KPS_effective (without diffusion correction) have been deleted, considering that these different types of uptake coefficients are compared and discussed in detail in Sect. 3.2 in the revised version. On the other hand, the upper and lower panel is labeled as A and B, respectively, and they are further explained in the text of Appendix C (Page 10, line 18) and in the figure caption. We believe these modifications can enhance its readability:

"**Appendix C**

**Comparison between KPS and CKD**

The KPS method is a recently developed analytical approximation method. The derivation of KPS is based on kinetic flux model framework and models describing interactions of gas species with aerosols in combination with the diffusion limitation theory for gas and particle uptake on a tube wall (Knopf et al., 2015, and references therein). This approximation method circumvents the complex operation procedures of previous numerical methods (e.g., the Brown and CKD methods), and therefore can be applied in a simpler way. As analyzed in KPS, the effective uptake coefficient $\gamma_{eff}$ can be experimentally determined as (Knopf et al., 2015):

$$\gamma_{eff} = \frac{d}{\omega \times t} \ln\left(\frac{C_0}{C}\right)$$

(C1)

where $d$ is flow tube diameter, $\omega$ is mean molecular speed of the gas reactant, $t$ is residence time of the gas reactant within the coated-wall region, $C_0$ and $C$ are gas reactant concentration at the flow tube inlet and outlet, respectively. After correction for gas molecular diffusion effects, the real uptake coefficient $\gamma$ is derived as follows:

$$\gamma = \frac{\gamma_{eff}}{1 - \gamma_{eff} \dfrac{3}{2 N_{Shw}^{eff} \times Kn}}$$

(C2)

in which $N_{Shw}^{eff}$ is the effective Sherwood number and $Kn$ is the Knudsen number, which can be expressed respectively as:

$$N_{Shw}^{eff} = 3.6568 + \frac{0.0978}{z^* + 0.0154} \quad \text{with} \quad z^* = L \times (\frac{\pi}{2}) \times \left(\frac{D}{F}\right)$$

(C3)

$$Kn = \frac{2\lambda}{d} \quad \text{with} \quad \lambda = \frac{3D}{\omega}$$

(C4)

where $z^*$ is dimensionless axial distance, $L$ is length of the coated-wall region, $D$ is molecular diffusion coefficient of the gas reactant within the carrier gas at experimental conditions, $F$ is volumetric flow rate of the fluid and $\lambda$ is mean free path of the gas reactant.

The CKD method in the present study is based on directly solving the differential equation, which is provided by Murphy and Fahey (1987) and used for description of the gas reactant concentration as a function of axial and radial position in a flow tube. Thus this CKD method can

possess higher accuracy than the previously used CKD interpolation method or the KPS method (Knopf et al., 2015;Li et al., 2016).

As shown in Fig. A2, with ideal laminar flow (i.e., without any local turbulence, LF regime) the KPS and CKD show perfect agreement for the derived uptake coefficient in the fractional loss range of 0.452 to 1 (shaded area in panel A). Due to the different algorithms employed, however, the CKD method (Murphy and Fahey, 1987;Cooney et al., 1974;Davis, 1973;Li et al., 2016) and the KPS method (Knopf et al., 2015) could derive contrasting uptake coefficient values when local turbulence occurs. If a fractional loss is larger than the critical fractional loss value (i.e., $1 - C/C_0 > 0.452$, in panel B), e.g., because of enhanced mass transport towards the coated-wall due to local turbulence, the KPS results in a negative uptake coefficient (blue dashed line in Fig. A2) while the CKD has no solution. From Eqn (C1), it can be found that an unrealistically high fractional loss can lead to a high $\gamma_{eff}$, which may cause a negative denominator in Eqn (C2) and therefore a derived negative uptake coefficient. For a fractional loss value smaller than 0, both methods derive negative uptake coefficients implying emissions of gas reactants from the coating (i.e., $C/C_0 > 1$, in panel B)."

[Figure]

**Figure A2.** Comparisons between uptake coefficients (derived from KPS and CKD methods, respectively) versus the fractional loss. Panel (A) displays the derived positive uptake coefficients under laminar flow (LF) regime, and panel (B) the derived negative ones due to emission (the left) or local turbulence effect (the right). For derivation of this plot, the specific experimental configuration is: gas reactant, $SO_2$; carrier gas, synthetic air; volumetric flow rate $F = 4$ L min$^{-1}$ at 1 atm and 296 K; flow tube dimension, d = 17 mm, L = 200 mm.

To improve the readability and clarity of the whole manuscript and to help the readers easily understand and further utilize our proposed method to evaluate their flow tube experiment results and to make better flow tube coating design, additional revisions have been made as shown below.

Some symbols and abbreviations in this manuscript have been changed or re-defined:

$\varepsilon_{max}$ (coating thickness) $\longrightarrow$ $\delta_g$ (geometric coating thickness) and $\delta_m$ (mass-based coating thickness);

$\varepsilon$ (roughness height) $\longrightarrow$ $\delta_r$ (roughness height);

$\gamma_{eff}$ (calculated effective uptake coefficient) $\longrightarrow$ $\gamma_{CKD}$, $\gamma_{KPS}$ (the uptake coefficient derived using the conventional CKD/KPS methods, respectively);

M-CKD (a modified CKD method) $\longrightarrow$ CKD-LT (a modified CKD method to account for local turbulence);

$C/C_0$ (concentration transmittance) $\longrightarrow$ $C/C_0$ (penetration);

Case 1 $\longrightarrow$ LT regime (local turbulence regime);

Case 2 $\longrightarrow$ LF regime (laminar flow regime);

All the symbols/abbreviations can be found in Table A.1 in Appendix A.

The additional revisions are marked with red color and the whole revised manuscript is shown below.

[revised manuscript text omitted]